# The Role of Apoptosis and Autophagy in the Hypothalamic-Pituitary-Adrenal (HPA) Axis after Traumatic Brain Injury (TBI)

**DOI:** 10.3390/ijms232415699

**Published:** 2022-12-10

**Authors:** Serpil Taheri, Züleyha Karaca, Ecmel Mehmetbeyoglu, Zuhal Hamurcu, Zeynep Yilmaz, Fatma Dal, Venhar Çınar, Halil Ulutabanca, Fatih Tanriverdi, Kursad Unluhizarci, Minoo Rassoulzadegan, Fahrettin Kelestimur

**Affiliations:** 1Department of Medical Biology, Medical Faculty, Erciyes University, 38030 Kayseri, Turkey; 2Betul-Ziya Eren Genome and Stem Cell Center, Erciyes University, 38030 Kayseri, Turkey; 3Department of Endocrinology, Medical Faculty, Erciyes University, 38030 Kayseri, Turkey; 4Department of Cancer and Genetics, Cardiff University, Cardiff CF10 3AT, UK; 5Department of Neurosurgery, Medical Faculty, Erciyes University, 38030 Kayseri, Turkey; 6INSERM-CNRS, Université de Nice, 06107 Nice, France; 7Department of Endocrinology, Medical Faculty, Yeditepe University, 34755 Istanbul, Turkey

**Keywords:** traumatic brain injury (TBI), inflammation, HPA axis, autophagy, apoptosis

## Abstract

Traumatic brain injury (TBI) is a major health problem affecting millions of people worldwide and leading to death or permanent damage. TBI affects the hypothalamic–pituitary–adrenal (HPA) axis either by primary injury to the hypothalamic–hypophyseal region or by secondary vascular damage, brain, and/or pituitary edema, vasospasm, and inflammation. Neuroendocrine dysfunctions after TBI have been clinically described in all hypothalamic–pituitary axes. We established a mild TBI (mTBI) in rats by using the controlled cortical impact (CCI) model. The hypothalamus, pituitary, and adrenals were collected in the acute (24 h) and chronic (30 days) groups after TBI, and we investigated transcripts and protein-related autophagy (*Lc3*, *Bcln1*, *P150*, *Ulk*, and *Atg5*) and apoptosis (pro-caspase-3, cleaved caspase-3). Transcripts related to autophagy were reduced in the hypothalamus, pituitary, and adrenals after TBI, however, this was not reflected in autophagy-related protein levels. In contrast, protein markers related to apoptosis increased in the adrenals during the acute phase and in the pituitary during the chronic phase. TBI stresses induce a variation of autophagy-related transcripts without modifying the levels of their proteins in the HPA axis. In contrast, protein markers related to apoptosis are increased in the acute phase in the adrenals, which could lead to impaired communication via the hypothalamus, pituitary, and adrenals. This may then explain the permanent pituitary damage with increased apoptosis and inflammation in the chronic phase. These results contribute to the elucidation of the mechanisms underlying endocrine dysfunctions such as pituitary and adrenal insufficiency that occur after TBI. Although the adrenals are not directly affected by TBI, we suggest that the role of the adrenals along with the hypothalamus and pituitary should not be ignored in the acute phase after TBI.

## 1. Introduction

The brain can be temporarily or permanently altered after a traumatic brain injury (TBI). First, a primary injury is defined as direct damage to the brain, including tissue shear and vascular damage [1,2,3]. A secondary injury may occur with an inflammatory response [4] depending on genetic predisposition [5]. Recently, we showed that certain polymorphisms of apolipoprotein-E (APO-E) synthesized after TBI in the central nervous system, including the hypothalamus–pituitary region, reduce the risk of pituitary insufficiency, and for this, one must take into account individual differences in treatment [5]. The hypothalamic–pituitary–adrenal (HPA) axis, which consists of the hypothalamus, pituitary, and adrenal glands, is responsible for providing an adequate cortisol response to maintain the homeostasis of organisms in stressful situations, such as in TBI [6]. Endocrine dysfunctions after TBI have been clinically described in all hypothalamic–pituitary axes [7,8,9]. In addition to frequently seen neuropsychiatric symptoms, one of the most important neuroendocrine disorders emerging after TBI is pituitary insufficiency [7,8,10,11,12]. Although these abnormalities that emerge after a TBI are temporary in most patients, they may be permanent in some [7].

The mechanism(s) of pituitary dysfunction, which can be seen in 27% of the survivors (6-36 months) after moderate to severe TBI, has/have not yet been fully elucidated [13]. The pituitary gland, together with the hypothalamus, is thought to act as the maestro for the synthesis and secretion of adenohypophyseal hormones (FSH and LH, TSH, GH, PRL, ACTH) and neurohypophyseal hormones (vasopressin and oxytocin). Hypopituitarism, a condition of pituitary hormone deficiency, results from impaired production of one or more anterior trophic hormones. Changes in pituitary functions after TBI result from the disruption of this hierarchical organization and may have global consequences in the short and long term. However, it is not clear where and how this hierarchical organization breaks down. Recent evidence suggests that neuroinflammation after TBI is responsible for many long-term clinical outcomes, including hypopituitarism. Many studies have shown increased secretion of TNF-α, an inflammatory marker from active astrocytes, and microglia in the brain after TBI [13]. Prolonged TNF-α production is harmful to neurons and leads to apoptosis and autophagy [14,15]. The role of autophagy and apoptosis in the emergence of HPA axis dysfunction after TBI is not yet known. 

Apoptosis and autophagy are important molecular processes that maintain the homeostasis of the organism and the cell, respectively. Although apoptosis fulfills its role by dismantling damaged or unwanted cells, autophagy maintains cellular homeostasis through recycling organelles and selective intracellular molecules. Apoptosis is defined as programmed cell death, and active cleaved caspase-3 has been shown to stimulate apoptosis in numerous studies of the process of apoptosis in mammals [16,17,18]. Basal levels of autophagy ensure the removal of abnormal proteins and aged or damaged organelles within the cell in the normal state. However, when the autophagy mechanism is over-activated, it can trigger apoptosis [19,20].

The controlled cortical impact (CCI) causes long-term dysregulation of the neuroendocrine stress response in the animal model and is mainly used to mimic human head trauma with different severity for developing preventative measures and reducing the impact of various types of primary injury [21]. 

It is already known that TNF-α increases in serum after TBI, and studies in animal models have focused on different regions of the brain [22]. Importantly, in tissues related to the HPA axis during the acute and chronic phases after TBI, the state of cells and tissues from the point of view of apoptosis or autophagy has not yet been studied. In this study, we sought to determine the presence of neuroinflammation after TBI in the HPA axis consisting of the hypothalamus, pituitary, and adrenal glands. The longer-term consequence of TBI is hypopituitarism, but it is unclear how, where, and when neuroinflammation triggers autophagy or apoptosis mechanisms. Assessment of autophagy and apoptosis in the HPA axis after TBI indicates disturbances with increased apoptosis in the adrenal glands preceding pituitary insufficiency. Here, the role of the adrenal glands after TBI is suggested.

## 2. Results

TBI was generated in rats by using the controlled cortical impact (CCI) model. The hypothalamus, pituitary, and adrenal tissues were removed in the acute (24 h) and chronic phase (30 days). 

### 2.1. mTBI-Induced Stress Increases TNF-α Protein Levels in the Adrenals

Generally, TNF-α protein levels are measured from the sera to detect inflammation in the HPA axis [21]. The HPA axis is activated acutely after a TBI as a result of the stress of the damage. In this study, we determined the TNF-α protein levels in a tissue-specific manner by Western blotting assay in order to show the effect of TBI only in the target tissues which was not previously determined in the HPA axis tissues. TNF-α protein levels were increased in the hypothalamus, pituitary, and adrenal tissues indicating inflammation in the acute and chronic phases (Figure 1A–C). TNF-α protein was not detectable in the hypothalamus of the control group (Figure 1A). 

### 2.2. mTBI Induces Alterations in the Transcript Levels of Autophagy Markers (Lc3, Bcln1, P150, Ulk, and Atg5) Distinctly in the Hypothalamus, Pituitary and Adrenals from the Acute Phase to the Chronic Phase

We investigated transcripts of autophagy *(Lc3, Bcln1, P150, Ulk*, and *Atg5*) in the hypothalamus, pituitary, and adrenal glands in the acute (24 h) and chronic (30 days) phases after TBI. Among the five transcripts, *P150* (dF = 2, *p* = 0.0004; dF = 2, *p* < 0.001) and Atg5 (dF = 2, *p* = 0.0012; dF = 2, *p* = 0.001) were significantly decreased from the acute phase and further decreased in the chronic phase after TBI in the hypothalamus (Figure 2A) and pituitary compared to the control group (Figure 2B). 

All other transcripts tested, the *Bcln1* transcript levels were mild, albeit not significantly lower in the hypothalamus and pituitary during the acute (respectively, dF = 2.2; *p* = 0.41 and *p* > 0.99) and chronic (respectively, dF = 2.2; *p* = 0.73 and *p* > 0.15) phases of TBI than the control group (Figure 2A,B). The *Ulk* transcript levels were unchanged in the hypothalamus in the acute (dF = 2; *p* = 0.27) and chronic phases (dF = 2; *p* = 0.43) of TBI (Figure 2A); however, in the pituitary during the acute phase *Ulk* transcripts were reduced (dF = 2; *p* = 0.049), which was not maintained during the chronic phase (dF = 2; *p* > 0.99) (Figure 2B). 

*Bcln1* was the only transcript to increase in the adrenals of the chronic group compared to the acute group of TBI and the control group (dF = 2, *p* = 0.007) (Figure 2C). Three other transcript levels (*Ulk, P150,* and *Atg5*) were significantly decreased compared to the control group in the acute phase of TBI (respectively, dF = 2, *p* = 0.011; dF = 2, *p* = 0.031; dF = 2, *p* = 0.023) (Figure 2C). The levels of the *Ulk* and *Atg5* transcripts were decreased compared to the control group in the acute and chronic phases of TBI. The decrease was significant in the adrenal only in the acute phase (*Ulk*: dF = 2, *p* = 0.025; *Atg5*: dF = 2, *p* = 0.023) (Figure 2C). 

When autophagy is induced, cytosolic Lc3-I is transformed to Lc3-II and localized in autophagosome membranes by the addition of PE. As a result, Lc3-II expression is widely considered to be a sign of autophagy induction [23]. When autophagy is induced, cytosolic Lc3-I is transformed to Lc3-II and localized in autophagosome membranes by the addition of PE. Both Beclin-1 and Lc3 proteins are crucial autophagy mediators and are involved in distinct phases of the autophagic process [24]. The Lc3-I protein was not transformed to the Lc3-II (Appendix A). Western blot analysis showed that Beclin-1 and Lc3 protein levels were similar in the hypothalamus, pituitary, and adrenal glands in the acute and chronic TBI groups compared to the control group (Figure 2D, Appendix A).

### 2.3. mTBI Increases Apoptosis First in the Acute Phase in the Adrenals and then in the Chronic Phase in the Pituitary

To evaluate the occurrence of apoptosis after TBI, in the acute and chronic phases, we assessed the expression of total caspase-3 and levels of Clv-caspase-3 (the active form of caspase 3) by using Western blot analysis in the HPA axis-related tissues. There was no detectable level of Clv-caspase-3 protein in the hypothalamus in the acute and chronic phases after TBI (Figure 3A). We found that the induction of Clv-caspase-3 showed a marked increase in the acute phase in the adrenal and in the chronic phase in the pituitary after TBI (Figure 3B,C). 

## 3. Discussion

Endocrine dysfunctions after TBI have been clinically described in all hypothalamic–pituitary axes. In addition, hypopituitarism was detected in 27% of survivors (6–36 months) after moderate to severe TBI. The HPA axis is responsible for providing an adequate cortisol level to maintain organisms’ homeostasis in the face of stress responses [9]. 

Based on the knowledge gained so far, it is tempting to assume that chronic inflammation plays a role in the development of long-term endocrine dysfunctions after TBI, especially in people with an inherited predisposition [5,13,25,26]. 

Previous studies have demonstrated that autophagy is activated after TBI, and researchers have hypothesized that inhibition of the autophagic pathway may ameliorate neurological deficits [27,28,29]. However, none of these studies focused on the HPA axis-related tissues that provide hormonal responses and, especially, an adequate cortisol response to maintain the organism’s homeostasis in stressful situations after TBI. 

Therefore, we investigated whether autophagy or apoptosis was involved in the regulation of cellular functions in the hypothalamic, pituitary, and adrenal tissues of rats after a mild controlled cortical impact (CCI) model and focused on transcription and protein markers associated with autophagy and apoptosis. The present study revealed the increased apoptosis ratio in the acute phase of TBI in the adrenal glands [30,31]. Remarkably, in the adrenal tissue, the expression of Clv-caspase-3 was increased in the acute phase, but it was not expressed in the chronic phase. Adrenal insufficiency due to suppression of HPA axis activation emerges in 11–13 percent of all TBI cases, based on clinical evidence [7]. 

According to our results, although transcription levels of autophagic markers showed significant changes in the hypothalamus, pituitary, and adrenal glands in acute and chronic phases after TBI, these changes were not reflected in protein levels of autophagy-related such as Lc3 and Beclin-1. Although transcript levels of autophagic markers were significantly lower in all tissues after TBI, especially during the chronic phase, these changes did not translate to tissue protein levels of these transcripts. The decrease in autophagic transcripts could indicate more general changes in levels either of pre- or posttranscriptional RNA regulation in the HPA axis after TBI. An in-depth study of the regulation of total transcripts during these phases should be planned soon.

The trauma-induced inflammatory response is a major component of TBI. This secondary injury after TBI occurs hours or weeks after injury and causes biochemical changes in proximal and distant tissues. The trauma not only causes neuroinflammation in the brain but also leads to a systemic inflammatory response. Proinflammatory cytokines play an important role in maintaining normal brain function and repair after TBI. However, the extreme and uncontrolled release of these cytokines, particularly interleukin (IL)-1β, IL-6, and TNF-α, can also cause a large amount of damage outside of the brain. Levels of these cytokines may increase thousands of times more than physiologically expected levels in the brain; however, that may not be reflected in the serum levels. TNF-α is an important inflammatory mediator that occurs after TBI [22]. As a result of our study, it was determined that TNF-α levels increased in the hypothalamus, pituitary, and adrenal in the acute and chronic phases after TBI. According to the literature, TNF-α levels were elevated in serum only in the acute period after TBI [32]. After a mild TBI, the increased results of TNF-α in the chronic period, especially in the adrenal, are quite remarkable.

Chronic inflammation is one of the most important triggers of apoptosis, defined as programmed cell death. Cleaved caspase-3 activates and stimulates apoptosis in many apoptotic processes in mammals. First, caspase-3 is synthesized as a 32 kilo Dalton (kDa) proenzyme, then it is cleaved into 12 and 17 k Da subunits. Two subunits (12 k Da and two 17 k Da) join to create the functionally active cleaved-caspase-3 enzyme. After the formation of the active cleaved caspase-3 enzyme, effector caspases, which include caspase-3, -6, and -7, initiate the degradation of many important cellular proteins. Finally, after degradation, the chromatin condenses and fragments the DNA [16,17,18].

Basal levels of autophagy ensure the removal of abnormal proteins and aged or damaged organelles within the cell in the normal state. However, when the autophagy mechanism is over-activated, it can trigger apoptosis. Autophagy-related (Atg) genes/proteins, including Beclin-1 *(Bcln1*) and microtubule-associated light chain-3 (LC3), play a major role and are frequently accepted as potential markers of autophagic activity [27,33]. Beclin-1 is involved in the very early stage of autophagosome formation and is considered an essential component for the initiation of autophagy. The LC3 protein has two forms, LC3-I and LC3-II. The LC3-I protein is localized in the cytoplasm under normal cell conditions. When the autophagy mechanism is triggered by various stresses, such as hypoglycemia, hypoxia, and inflammation, a cytosolic form of LC3-I is transformed into LC3-II by conjugation of a lipid molecule called phosphatidyl-ethanolamine (PE) for incorporation into the membrane of autophagosomes [33,34,35]. P150 is the largest component of the dynein–dynactin complex, which has an indispensable role in autophagy. Autophagy-activating kinase Unc-51-like (ULK)1 plays a critical role in regulating autophagy initiation independent of (Atg) 5 and Atg7. Many studies have shown that increasing autophagy upregulates ULK1 expression [36,37].

At this point, increased apoptotic or autophagic activity associated with increased neuroinflammation in tissues associated with the HPA axis after TBI may lead to loss of secretory or regulatory cells, which may be the cause of pituitary deficiency occurring after TBI. There are no studies in the literature showing how autophagy or apoptosis mechanisms play a role in the HPA axis after TBI to date. This knowledge could help in the development of new targeted treatment strategies.

In 2011, Chen et al. created TBI in rats by using the fluid percussion injury model and found that the rate of apoptosis increased in the hypothalamus and pituitary in the 7 and 14 days after TBI [38]. Tan et al. developed a model of intracranial hypertension to investigate its effects on the HPA axis caused by TBI. They found that 24-h intracranial hypertension increased the rate of apoptosis, particularly in the hypothalamus and pituitary [16]. However, we found that the ratio of apoptosis markers in the hypothalamus and pituitary did not change in the acute phase after TBI. In contrast, we found increased expression of the apoptosis marker Clv-Caspase-3 in the adrenal glands as a target organ in the acute phase of the HPA axis. In the pituitary, here, the apoptosis expression marker “Clv-caspase-3” was only detected in the chronic phase after TBI. However, the differences between the results of studies investigating the effects of TBI on the HPA axis may also be due to the different TBI models used.

Finally, our results suggest that adrenal glands in the acute phase after TBI may play an important role in the mechanism of post-TBI-emergent endocrine dysfunctions, particularly pituitary insufficiency. We suggest that increased apoptosis of the adrenal glands in the acute phase can also impair communication along the HPA axis, and disruption of communication leads to permanent damage to the pituitary in the chronic phase. In addition to all this, these results show that the adrenal glands may play an important role in the mechanism of diseases other than pituitary insufficiency that occurs after TBI. 

Although the adrenal glands are not directly affected by TBI, we suggest that adrenal glands contribute to the endocrine dysfunctions occurring after TBI.

## 4. Materials and Methods

### 4.1. Animals

A total of 42 healthy eight-week-old adult male Sprague–Dawley rats weighing 250–300 g, were used in this study. These rats were housed under a 12 h light/dark cycle at 23 °C and allowed ad libitum access to food and Erciyes University, Genome and Stem Cell Center (GenKok), Transgenic Department, Kayseri, Turkey. 

### 4.2. Experimental Design

The male rats were randomly divided into three groups, and each group included 14 rats. TBI was generated in rats by using the CCI model [21,39,40]. The details were as follows: (1) control group, no operated group (only opened 10 mm median linear incision was then sutured under anesthesia); (2) acute TBI, rats were sacrificed at 24 h after TBI; (3) chronic TBI, rats were sacrificed 30 days after TBI. mRNA expression analyses were performed on 14 rats for each group. Western blot analyses were performed on nine rats for each group because enough protein was not obtained (excluded sample numbers are indicated on the original Western blot images). All of the findings were derived from the entire total tissue of the hypothalamus, pituitary, and adrenals [30]. All experiments were performed on randomized and controlled double-blind. Tissue samples were stored in Trizol (Thermo Fisher Scientific, Waltham, MA, USA) at −80 °C until use. The experimental design is provided in Figure 4.

### 4.3. Models of TBI

The CCI techniques, parameters, and postoperative care have all been thoroughly reported previously [21,39,40]. Rats were sedated with isoflurane (2.0–2.5 percent in 100% O_2_, 2.0 mL/min flow rate) and put in a stereotaxic frame with the head in a horizontal plane about the interaural line. The spontaneous respiration of the rats was not suppressed during the experiment. Depth of anesthesia was monitored by the chin and skeletal muscle tone. Every surgical procedure was carried out in an aseptic environment. In this method, a craniotomy is performed, and head trauma is created in the area of interest with a computerized pneumatic system at the desired speed, depth, and duration. After anesthesia, the scalp of the rat, which was fixed in the prone position with a standard nailed frame, was shaved, and the area was cleaned with 10% povidone-iodine, and the periosteum was dissected by cutting the skin and fascia with a 10-mm median linear incision. The skin and galea were fixed to the laterals with a bulldog clip to provide an adequate visual field. Then, in the left parasagittal region, a craniectomy defect was produced by drilling the cranium with a 3-mm diameter dental drill 2 mm lateral to the sagittal suture and 2 mm posterior to the bregma. By varying the velocity with which the exposed dura was struck and the depth of tissue compression, a mild injury was induced. The TBI model had the following parameters: high pressure of 200 Kpa, and depth of 1.0 mm. This model is referred to as a mild-TBI model in the literature [41]. Finally, the scalp was sutured with 3/0 silk and the wound was cleaned again with 10% povidone-iodine. No operation was performed on the control group and only the scalp of the rat was open 10 mm median linear incisions then the scalp was sutured. Following the injury, the animals were observed daily. Rats that showed signs of pain (frozen, stooped posture, or vocalization) or infection (swelling, redness, or discharge) were eliminated from the experiment, as were those that lost more than 20% of their body weight [21,40].

### 4.4. RNA Isolation and Real-Time PCR

Total RNA isolation was made from tissues using Trizol (Thermo Fisher Scientific, MA, USA), and cDNA was synthesized from the RNA samples by using an Evo Script cDNA synthesis kit (Roche, Mannheim, Germany). The cDNA procedure was conducted according to the manufacturer’s protocol and samples were quantified by using a Roche 480 real-time PCR (Roche, Mannheim, Germany). Transcript levels of *Ulk, P150, Atg5,* and *Bcln1* genes in the hypothalamus–pituitary–adrenal of all groups were determined. The *Actb* gene was used as a housekeeping gene. All PCR experiments were repeated twice. The Ct values were normalized by using the $2^{-\Delta \Delta \mathrm{Ct}}$ method [30,42].

### 4.5. Western Blot Analysis

Standard procedures were used to extract total proteins from dissected hypothalamus, pituitary, and adrenal glands [43,44]. A detergent-compatible protein assay kit was used to quantify the total protein concentration in each sample (DC kit; Bio-Rad, Hercules, CA, USA). For protein separation, aliquots comprising 40 µg of total protein from each sample were electrotransferred to polyvinylidene difluoride membranes by using sodium dodecyl sulfate (SDS)-polyacrylamide gel electrophoresis with a 4–20% gradient. The membranes were blocked with a blocking buffer (TBS-T) (0.1 Triton X-100 in Tris-buffered saline–Tween 20) for 60 min. The membranes were probed with the primary antibodies listed after being washed with TBS-T: Tnf-α (Proteintech, Catalog no: 17590-1-AP, Rosemont, IL, USA), Lc3 (Cell Signaling, Catalog no:2775S, Danvers, MA, USA), Beclin-1(Cell Signaling, Catalog no:3738S, Danvers, MA, USA), pro-caspase-3 (Cell Signaling, Catalog no:9662S, Danvers, MA, USA), cleaved-caspase-3 (Cell Signaling, Catalog no:9661S, Danvers, MA, USA), and β-actin (Proteintech, Catalog no:60008-1-Ig, Rosemont, IL, USA). The membranes were treated with horseradish peroxidase-conjugated anti-rabbit (Bio-Rad, catalog no:170-6515, Hercules, CA, USA) or anti-mouse secondary antibody (Bio-Rad, Catalog no:170-6516, Hercules, CA, USA) after being rinsed with TBS-T. TBS-T containing 5% dry milk was used to dilute all antibodies. Clarity Western ECL substrate (Bio-Rad, Hercules, CA, USA) was used for chemiluminescence detection, and the blots were viewed with a Chemidoc MP Imaging System (Bio-Rad, Hercules, CA, USA). The membranes were incubated with primary antibodies overnight at 4 °C, then with secondary antibodies for 1 h. Chemiluminescence detection was performed with Clarity Western ECL substrate (Bio-Rad, Hercules, CA, USA) and the blots were visualized with a Chemidoc MP Imaging System and quantified with a densitometer by using the imager application program. All Western blot experiments were independently repeated at least twice. All of the antibodies were validated and made ready for use by the supplier company “according to the antibody validation principle of Uhlen et al. (see manufacturer’s website for antibody validation princible: https://www.cellsignal.com/about-us/cst-antibody-validation-principles) (accessed 7 December 2022) [43,44,45,46,47].

### 4.6. Statistical Analysis

The compliance of the data to normal distribution was evaluated by the histogram, q-q graphs, and Shapiro–Wilk test. For determination of the differences in gene expression data, independent samples were analyzed by ordinary one-way ANOVA or by Kruskal–Wallis test according to their distribution. If samples had normal distribution, parametric ordinary one-way ANOVA test was used with Tukey post-hoc test. If samples did not have normal distribution non-parametric Kruskal–Wallis test with Dunnett’s post hoc test was used. 

Differences in protein expression were analyzed by unpaired Student t-test or by Mann–Whitney-U test according to their distribution. If samples had normal distribution, a parametric unpaired Student t-test was used. If samples did not have normal distribution, a nonparametric Mann–Whitney-U test was used. 

Graph-Pad Prism 8 packages were used for statistical analysis. Results with *p* values <0.05 were considered statistically significant. All data are expressed as the mean with SD. The sample size was calculated with power analysis, the amount of Type I error (alpha) was 0.05, the power of the test (1-beta) was 0.95, and the effect size was 0.82 while using the independent samples one-way ANOVA test for minimum sample size required a significant difference of nine for each group; a total of 27 male rat was used for G-power (v3.1) power analysis.

## 5. Conclusions 

In this study, we sought to determine which, where, and when the autophagy or apoptosis mechanisms in the HPA axis of neuroinflammation occur after TBI trigger and its relationship with endocrine dysfunctions that occur after TBI. Although the adrenal glands are not directly affected by TBI, we suggest that adrenal glands contribute to the endocrine dysfunctions occurring after TBI.

## Figures and Tables

**Figure 1 ijms-23-15699-f001:**
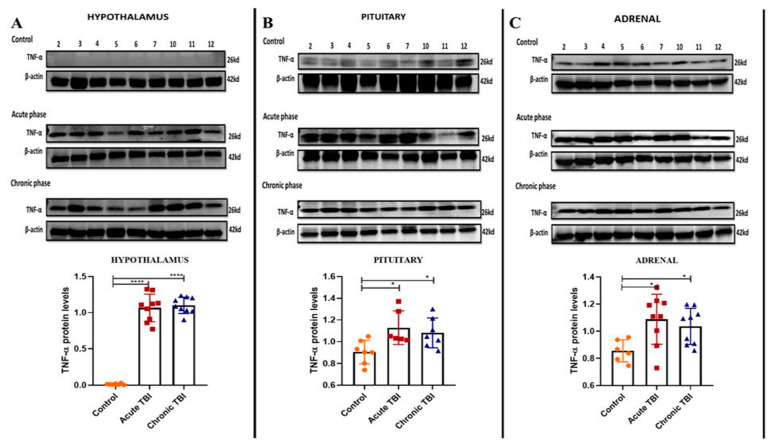
Western blot images and bar graphs of Tnf-α levels in the acute and chronic phase after TBI in the hypothalamus, pituitary, and adrenals. (**A**) Tnf-α protein levels in the hypothalamus after TBI in the acute, chronic and control groups. (**B**) Tnf-α protein levels in the pituitary after TBI in the acute, chronic and control groups. (**C**) Tnf-α levels in the adrenals after TBI in the acute, chronic and control groups. β-Actin was used as a loading control. Data are presented as mean ± SD (n = 9 rats in each group). (* *p* < 0.05, **** *p* < 0.0001). All experiments were independently repeated at least twice.

**Figure 2 ijms-23-15699-f002:**
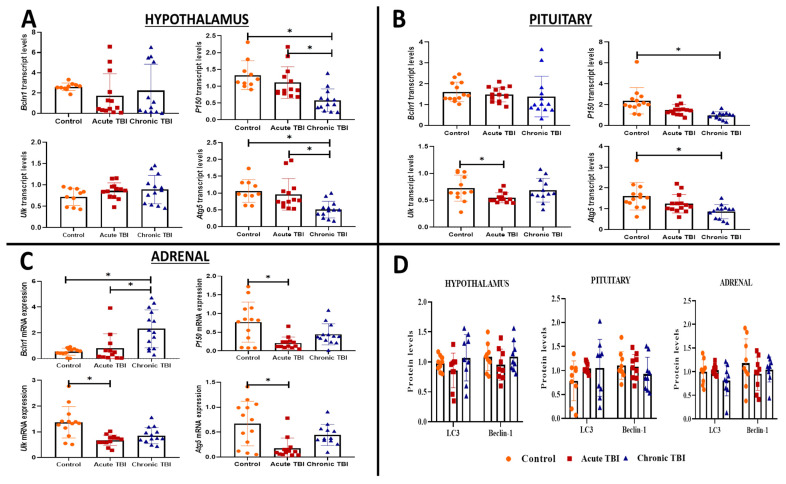
The transcript levels of *Bcln1, P150, Ulk*, and *Atg5* genes and protein levels of LC-3, Beclin-1 in the hypothalamus, pituitary, and adrenals in the acute and chronic phases after TBI. (**A**) The transcript levels of *Bcln1, P150, Ulk,* and *Atg5* genes in the hypothalamus after TBI in the acute, chronic, and control groups (**A**–**D**) (Bar Graph) (* *p* < 0.05). Box plots are expressed as mean ± SD (n = 14 rats in each group). (**B**) The transcript levels of *Bcln1, P150, Ulk,* and *Atg5* genes in the pituitary after TBI in the acute, chronic and control groups (**A**–**D**) (Bar Graph) (* *p* < 0.05). Box plots are as mean ± SD (n = 14 rats in each group). (**C**) The transcript levels of *Bcln1, P150, Ulk,* and *Atg5* genes in the adrenals after TBI in the acute, chronic and control groups (**A**–**D**) (Bar Graph) (* *p* < 0.05). Box plots are expressed as mean ± SD (n = 14 rats in each group). (**D**) The protein levels LC-3 and Beclin-1 protein levels in the hypothalamus, pituitary, and adrenals after TBI in the acute, chronic, and control groups. Data are presented as mean ± SD (n = 9 rats in each group). All experiments were independently repeated at least twice.

**Figure 3 ijms-23-15699-f003:**
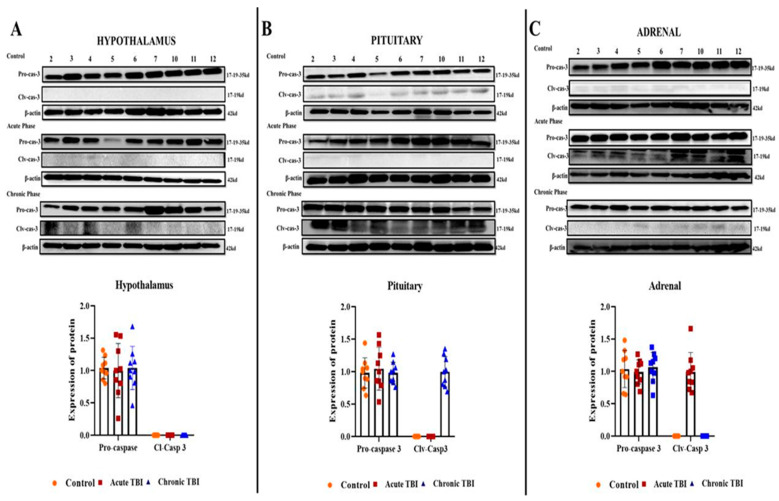
Western blot images of Clv-caspase-3 in the acute and chronic phase after TBI in the hypothalamus, pituitary, and adrenals. (**A**) Clv-caspase-3 protein levels in the hypothalamus after TBI in the acute, chronic, and control groups. (**B**) Clv-caspase-3 protein levels in the pituitary after TBI in the acute, chronic, and control groups. (**C**) Clv-caspase-3 protein levels in the adrenals after TBI in the acute, chronic, and control groups. β-Actin was used as a loading control (n = 9 rats in each group). All experiments were independently repeated at least twice.

**Figure 4 ijms-23-15699-f004:**
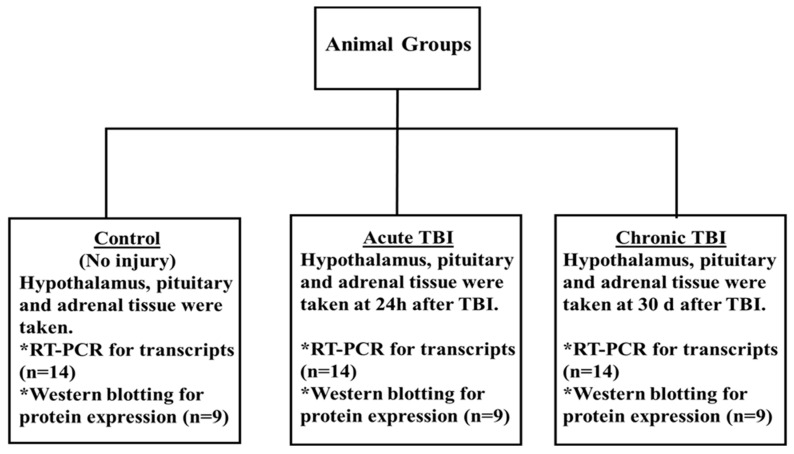
Experimental design. Anesthetized male rats were subjected to TBI with the CCI. The control group was sacrificed directly without any operation. The acute TBI group was sacrificed at 24 h after TBI with CCI. The chronic TBI group animals were sacrificed 30 days after TBI with CCI. The hypothalamus, pituitary, and adrenals were taken from each group. mRNA expression analyses were performed on 14 rats for each group. Western blot analyses were performed on nine rats for each group. All of the findings were derived from the entire total tissue of the hypothalamus, pituitary, and adrenals. All PCR and Western blot analyses for detecting transcript and protein levels were repeated at least twice independently and blindly.

## Data Availability

All analyzed data during this study are included in this published article. The raw data of the study are available from the corresponding author on request.

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
