# Peer review of "The Role of Apoptosis and Autophagy in the Hypothalamic-Pituitary-Adrenal (HPA) Axis after Traumatic Brain Injury (TBI)"

_ijms, 2022, doi:10.3390/ijms232415699_

Round 1
Reviewer 1 Report
In this manuscript, the authors investigate the effects of TBI on the HPA axis, specifically autophagy and apopotosis, using a cortical impact model of TBI. The manuscript needs improvement in several areas.
Major:
1) The introduction requires significant revision. Most of the information, especially details on autophagy and apoptotic signaling, would likely be better suited to the discussion. The introduction does not appropriately setup the goals of the study, which was to investigate the role of these processes in TBI-induced dysregulation of the HPA axis. The authors should provide more of a description of what has been done/is currently known regarding HPA dysfunction in TBI and highlight where the current study fills in the knowledge gaps in the literature. Several seemingly relevant publications are not referenced or discussed in the manuscript, a few of which are highlighted below:
Bromberg, Caitlin E et al. “Sex-Dependent Pathology in the HPA Axis at a Sub-acute Period After Experimental Traumatic Brain Injury.” Frontiers in neurology vol. 11 946. 30 Sep. 2020
Tapp, Zoe M et al. “A Tilted Axis: Maladaptive Inflammation and HPA Axis Dysfunction Contribute to Consequences of TBI.” Frontiers in neurology vol. 10 345. 24 Apr. 2019
Kosari-Nasab, Morteza et al. “The blockade of corticotropin-releasing factor 1 receptor attenuates anxiety-related symptoms and hypothalamus-pituitary-adrenal axis reactivity in mice with mild traumatic brain injury.” Behavioural pharmacology vol. 30,2 and 3-Spec Issue (2019): 220-228.
Russell, Ashley L et al. “Differential Responses of the HPA Axis to Mild Blast Traumatic Brain Injury in Male and Female Mice.” Endocrinology vol. 159,6 (2018): 2363-2375.
2) Please clarify whether the CCI model used produces a mild TBI or a severe TBI. The authors state in the introduction that CCI "is mainly used to mimic human severe head trauma" but then state in the Methods that the produce a mild CCI injury. Moreover, the authors state that the injury produced by the CCI model can be controlled by varying impact velocity and deformation depth, however, provide pressure and depth measures. While the authors state that this is a mild injury, a depression depth of 1.5 mm has been cited as a severe TBI (Romine et al., 2014 "Controlled cortical impact model for traumatic brain injury." Journal of visualized experiments: JoVE, 90 e51781).
3) The authors should consider the impact that the TBI procedure, largely the surgery itself, could have on the expression of inflammatory mediators in the brain. It is well established that peripheral inflammation can cause and impact neuroinflammation. As such, please clarify the details of the control group. It is defined as "no injury" and as "no operation." Are these non-handled controls or shams (i.e. surgery but no CCI)? If the control group is non-handled controls, the authors should addi a sham control group where the rats undergo all aspects of anesthesia, surgery, bore hole drilling without the CCI. This would control for any contribution of the surgery to the effects observed with the CCI.
4) The authors should include bar graphs for the caspase 3 data presented in Figure 4, similar as to what is presented in Figure 2.
5) The discussion requires some improvement. The authors fail to discuss or address the data presented for TNF-alpha. Considering that the authors highlight that TNF-alpha has largely been measured in the serum, as opposed to in the brain, this seems like a finding worth discussing.
Author Response
Reviewer 1:
Major:
1) The introduction requires significant revision. Most of the information, especially details on autophagy and apoptotic signaling, would likely be better suited to the discussion. The introduction does not appropriately setup the goals of the study, which was to investigate the role of these processes in TBI-induced dysregulation of the HPA axis. The authors should provide more of a description of what has been done/is currently known regarding HPA dysfunction in TBI and highlight where the current study fills in the knowledge gaps in the literature. Several seemingly relevant publications are not referenced or discussed in the manuscript, a few of which are highlighted below:
Bromberg, Caitlin E et al. “Sex-Dependent Pathology in the HPA Axis at a Sub-acute Period After Experimental Traumatic Brain Injury.” Frontiers in neurology vol. 11 946. 30 Sep. 2020
Tapp, Zoe M et al. “A Tilted Axis: Maladaptive Inflammation and HPA Axis Dysfunction Contribute to Consequences of TBI.” Frontiers in neurology vol. 10 345. 24 Apr. 2019
Kosari-Nasab, Morteza et al. “The blockade of corticotropin-releasing factor 1 receptor attenuates anxiety-related symptoms and hypothalamus-pituitary-adrenal axis reactivity in mice with mild traumatic brain injury.” Behavioural pharmacology vol. 30,2 and 3-Spec Issue (2019): 220-228.
Russell, Ashley L et al. “Differential Responses of the HPA Axis to Mild Blast Traumatic Brain Injury in Male and Female Mice.” Endocrinology vol. 159,6 (2018): 2363-2375.
Answer: The introduction has been revised and relevant articles were cited.
2) Please clarify whether the CCI model used produces a mild TBI or a severe TBI. The authors state in the introduction that CCI "is mainly used to mimic human severe head trauma" but then state in the Methods that the produce a mild CCI injury. Moreover, the authors state that the injury produced by the CCI model can be controlled by varying impact velocity and deformation depth, however, provide pressure and depth measures. While the authors state that this is a mild injury, a depression depth of 1.5 mm has been cited as a severe TBI (Romine et al., 2014 "Controlled cortical impact model for traumatic brain injury." Journal of visualized experiments: JoVE, 90 e51781).
Answer: We apologize for the mistake in this section. The error has been corrected and the relevant reference has been added.
3) The authors should consider the impact that the TBI procedure, largely the surgery itself, could have on the expression of inflammatory mediators in the brain. It is well-established that peripheral inflammation can cause and impact neuroinflammation. As such, please clarify the details of the control group. It is defined as "no injury" and as "no operation." Are these non-handled controls or shams (i.e. surgery but no CCI)? If the control group is non-handled controls, the authors should add a sham control group where the rats undergo all aspects of anesthesia, surgery, and borehole drilling without the CCI. This would control for any contribution of the surgery to the effects observed with the CCI.
Answer: No operation was performed on the control group, only under anesthesia was the scalp of the rat open through 10 mm median linear incisions and then the scalp was sutured.
4) The authors should include bar graphs for the caspase 3 data presented in Figure 4, similar as to what is presented in Figure 2.
Answer: The pro-caspase-3 and caspase-3 bar graphs are added in Figure 4.
5) The discussion requires some improvement. The authors fail to discuss or address the data presented for TNF-alpha. Considering that the authors highlight that TNF-alpha has largely been measured in the serum, as opposed to in the brain, this seems like a finding worth discussing.
Answer: The reason we haven't included much discussion about TNF-α is that our main focus is on the mechanisms of apoptosis and autophagy in the HPA axis after TBI associated with hypopituitarism. However, the discussion was developed based on the reviewer's suggestions and additional information was added about TNF-α the discussion.
Reviewer 2 Report
The Role of Apoptosis and Autophagy in the Hypothalamic-Pituitary-Adrenal (HPA) Axis after Traumatic Brain Injury (TBI) manuscript provides experimental results related to endocrine dysfunctions after TBI connected to the HPA axis in a logical and convincing manner, presenting data regarding autophagy and apoptosis. The regulation of autophagy and apoptosis in HPA axis-related tissues in the acute and chronic phases after TBI has not yet been demonstrated. Western blot images are of good quality and the statistical analysis is appropriate. The authors underline that adrenals contribute to the endocrine disturbances emerging after TBI.
My suggestions for improvement of the manuscript are:
1. a further explanation involving homeostasis mechanisms for autophagy and apoptosis after TBI in connection with the HPA axis;
2. development of the paragraph: ”it's tempting to speculate that chronic inflammation plays a role in the development of long-term endocrine dysfunctions following TBI, especially in people who have a hereditary susceptibility” - it seems that there are necessary supplementary scientific data;
3. further study directions with relevance to clinical practice - targets for therapeutic interventions
4. In the discussion, there are no considerations about TNF-α, but only about autophagy and apoptosis.
5. Finally, looking for conclusions about the role of apoptosis and autophagy in the HPA axis after TBI, it seems that this part is missing, and we have only the role of adrenals that contribute to the endocrine disturbances emerging after TBI.
Author Response
Reviewer 2: Comments and Suggestions for Authors
The Role of Apoptosis and Autophagy in the Hypothalamic-Pituitary-Adrenal (HPA) Axis after Traumatic Brain Injury (TBI) manuscript provides experimental results related to endocrine dysfunctions after TBI connected to the HPA axis in a logical and convincing manner, presenting data regarding autophagy and apoptosis. The regulation of autophagy and apoptosis in HPA axis-related tissues in the acute and chronic phases after TBI has not yet been demonstrated. Western blot images are of good quality and the statistical analysis is appropriate. The authors underline that adrenals contribute to the endocrine disturbances emerging after TBI.
My suggestions for improvement of the manuscript are:
- a further explanation involving homeostasis mechanisms for autophagy and apoptosis after TBI in connection with the HPA axis;
Answer: In line with the referee's suggestion, necessary additions were made to the discussion section of the manuscript.
- development of the paragraph: ”it's tempting to speculate that chronic inflammation plays a role in the development of long-term endocrine dysfunctions following TBI, especially in people who have a hereditary susceptibility” - it seems that there are necessary supplementary scientific data;
Answer: In line with the referee's suggestion, necessary additions were made to the introduction section of the manuscript.
- further study directions with relevance to clinical practice - targets for therapeutic interventions
Answer: In line with the referee's suggestion, necessary additions were made to the discussion section of the manuscript.
- In the discussion, there are no considerations about TNF-α, but only about autophagy and apoptosis.
Answer: The discussion was developed based on the reviewer's suggestions and additional information was added about TNF-α the discussion.
- Finally, looking for conclusions about the role of apoptosis and autophagy in the HPA axis after TBI, it seems that this part is missing, and we have only the role of adrenals that contribute to the endocrine disturbances emerging after TBI.
Answer: In line with the referee's suggestion, necessary additions were made to the discussion section of the manuscript.
Round 2
Reviewer 1 Report
I thank the authors for their responses to the earlier comments and significant revision of the manuscript. I feel that these changes have largely improved the manuscript. I have some additional minor revisions suggested below:
1) These statements made in the introduction (2nd paragraph) are lacking references: "Recent evidence suggests that neuroinflammation after TBI is responsible for many long-term clinical outcomes, including hypopituitarism. Many studies have shown increased secretion of TNF-α, an inflammatory marker from active astrocytes, and microglia in the brain after TBI."
2) From the final paragraph of the introduction... "...apoptosis or autophagy did not yet been studied." should read "has not yet been studied"
3) The authors should confirm that they have uploaded the updated version of Figure 4. The bar graphs for Figure 4 (caspase) are not included in the peer-review version of the manuscript.
4) Please clarify the meaning of this sentence from the discussion (6th paragraph): "Levels of these cytokines may increase thousands of times more than physiologically expected levels in brain, however that could be find in the serum."
5) In discussing the results of other studies investigating the effects on HPA following TBI, the authors should consider or mention that the varying results may result from the use of different models of TBI. Some impact models produce more movement of the brain within the skull, which could impact the results. Also, the area chosen for TBI can impact what regions of the brain may be affected in the short and long term.
6) The final paragraph of the discussion requires some language editing.
Author Response
I thank the reviewers for their efforts. The changes requested by reviewer 1 have been made to the manuscript and are also listed below.
1) These statements made in the introduction (2nd paragraph) are lacking references: "Recent evidence suggests that neuroinflammation after TBI is responsible for many long-term clinical outcomes, including hypopituitarism. Many studies have shown increased secretion of TNF-α, an inflammatory marker from active astrocytes, and microglia in the brain after TBI."
Answer: The relevant reference was added.
2) From the final paragraph of the introduction... "...apoptosis or autophagy did not yet been studied." should read "has not yet been studied"
Answer: The sentence was corrected.
3) The authors should confirm that they have uploaded the updated version of Figure 4. The bar graphs for Figure 4 (caspase) are not included in the peer-review version of the manuscript.
Answer: New figure 4 was added in the manuscript.
4) Please clarify the meaning of this sentence from the discussion (6th paragraph): "Levels of these cytokines may increase thousands of times more than physiologically expected levels in brain, however that could be find in the serum."
Answer: The sentence was corrected as” Levels of these cytokines may increase thousands of times more than physiologically expected levels in the brain, however, that may not be reflected in the serum levels.”
5) In discussing the results of other studies investigating the effects on HPA following TBI, the authors should consider or mention that the varying results may result from the use of different models of TBI. Some impact models produce more movement of the brain within the skull, which could impact the results. Also, the area chosen for TBI can impact what regions of the brain may be affected in the short and long term.
Answer: The sentence “However, the differences between the results of studies investigating the effects of TBI on the HPA axis may also be due to the different TBI models used.” was added at the end of the paragraph.
6) The final paragraph of the discussion requires some language editing.
Answer: The final paragraph was revised.
Reviewer 2 Report
The authors made sufficient improvements to the manuscript. They adequately addressed all my previous comments and suggestions so that in this step, I do not have further comments, and I conclude for acceptance in the present form.
Author Response
Dear Reviewer
Thank you very much for your efforts
Best regards